

# Regulation of fruit quality formation in strawberry: from omics to biotechnology

Zhang-Ying Wang[1], An-Qing Shen[1], Yan-Xin Ge[1], Cheng-Ling Zhou[1], Yu-Shan Qiao[2], Ai-Sheng Xiong[3] and Guang-Long Wang[1]

[1] School of Life Science and Food Engineering, Huaiyin Institute of Technology, Huaian, China
[2] Institute of Pomology, Jiangsu Academy of Agricultural Sciences, Nanjing, China
[3] State Key Laboratory of Crop Genetics & Germplasm Enhancement and Utilization, College of Horticulture, Nanjing Agricultural University, Nanjing, China

## ABSTRACT

Strawberry (*Fragaria × ananassa* Duch.), a popular fruit, is well known for its bright color, unique flavor, and high nutritional value. The quality of strawberries will greatly affect consumers' choices, market share, and farmers' profits. The formation of strawberry quality is a complex biological process involving the interaction of multiple genetic factors and environmental conditions. In recent years, there has been great progress on investigating strawberry quality formation and regulation in the world. In this review, we summarized the factors from internal to external that affect strawberry formation, and focused on the application of omics technologies such as genomics, transcriptomics, proteomics, and metabolomics in the study of strawberry quality, as well as the potential of modern technologies in quality improvement. The viewpoints in this article may provide new ideas for breeders and scientists aimed to regulate and improve strawberry quality in the future.

## INTRODUCTION

Strawberry (*Fragaria × ananassa* Duch.) belongs to the Rosaceae family. It has been cultivated in Europe since the 18th century, and is a hybrid of *Fragaria chiloensis* (Chilean strawberry) and *Fragaria virginiana* (Virginia strawberry, native to the eastern United States). Strawberries are widely cultivated and consumed worldwide due to their charming appearance, good taste, nutrition, and health promoting properties, making them a fruit with great economic value (*Song et al., 2023*). The quality of strawberry fruits not only affects consumer acceptance, but also determines their market value and economic benefits. The quality characteristics of strawberries include taste, color, hardness, total soluble solids, volatile compounds, acidity, nutritional value, and storage resistance, which directly contribute to consumer acceptance and satisfaction. With the development of molecular biology technology, the regulatory mechanism of strawberry quality formation has gradually been revealed, providing scientific basis for the improvement and optimization of strawberry quality. This article will review the regulatory mechanisms of strawberry quality formation, and explore the application of modern technology and omics in strawberry

Corresponding authors
Guang-Long Wang,
wanggl89@hyit.edu.cn
Yu-Shan Qiao ,
qiaoyushan@njau.edu.cn

quality regulation. The results from the current work will provide new ideas for breeding experts and researchers committed to high-quality strawberry.

## SURVEY METHODOLOGY

We searched key words "strawberry/fruit quality/omics/postharvest preservation/biotechnology/nutritional quality" in combination with the subject title or its free term, respectively, in PubMed (https://pubmed.ncbi.nlm.nih.gov/) and Web of Science (https://webofscience.clarivate.cn/wos/alldb/basic-search). The literature reviews were mainly classified into three parts: factors affecting strawberry quality formation, omics used in strawberry quality, and technology in strawberry quality regulation.

## FACTORS AFFECTING STRAWBERRY QUALITY FORMATION

### Variety

The genotype of strawberries is a congenital factor that determines their quality. Different varieties have significant differences in fruit size, color, texture, sweetness, phenolic compounds, volatiles, and disease resistance. It was reported that 'Sabrina' had a solid firmness, whereas a relatively high total soluble solids content was observed in 'Candonga' fruit throughout several consecutive cropping seasons (*Cervantes et al., 2020*). Therefore, choosing suitable varieties is crucial for meeting specific market demands and improving market competitiveness for strawberry production.

### Developmental stage

The coloration, metabolite accumulation, and flavor profile of strawberry fruits undergo remarkable transformations during their distinct developmental stages, which exert significant influences on the formation of their ultimate quality attributes. Over the period of ripening, strawberries exhibit a dynamic chromatic evolution, transitioning through distinct color phases. This process is closely related to the accumulation of anthocyanins, carotenoids, and chlorophyll, which not only contribute to the fruit color, but also their antioxidant properties and nutritional values (*Yue et al., 2024*). It is well recognized that strawberry fruit undergoes a significant increase in anthocyanin content during fruit ripening (*Song et al., 2015*). During natural ripening, the activation of the phenylpropanoid biosynthesis and alpha-linolenic acid metabolic pathways was responsible for formation of aroma compounds in creamy strawberry cultivars, whereas terpenes were the major volatiles in aroma-free strawberry cultivars (*Fang et al., 2024*). Ripening 'Benihoppe' fruits accumulated more sucrose and citric acid with a little cyanidin and higher firmness, whereas more fructose, glucose, malic acid, and ascorbic acid were observed during 'Fenyu No.1' fruit development (*Yang et al., 2024*). With the development of strawberry fruit, the deposition of cuticle and wax on a unit surface-area basis decreases, accounting for their high susceptibility to the disorders of water soaking and cracking (*Straube et al., 2024*). Therefore, It is generally believed that sugar, aroma, and anthocyanins are usually increased during strawberry fruit development.

## Storage condition

Strawberries are prone to postharvest quality deterioration due to their high water content, soft texture, and active metabolic activity, which can lead to issues such as weight loss, texture softening, color changes, and nutrient loss. During postharvest storage process, weight loss, decay incidence, decreased fruit skin brightness, and reduced soluble protein were determined in strawberry fruits (*Zhang et al., 2022a*). In another study, strawberry fruits were found to undergo continuous softening with cell wall component degradation (*Wang et al., 2025*). Similarly, reduction in firmness, total soluble solids, total phenolics content, total flavonoid content, and ascorbate content was detected during storage, which was delayed by resveratrol application (*Fan et al., 2022a*). Generally, postharvest strawberries exhibit rapid decline of water, accompanied by accelerated ethylene release and cell wall decomposition, which result in increasing ripening and softening.

## Biotic stress

Strawberries are susceptible to various diseases during their growth process, which can have a serious impact on their growth, development, and fruit quality. For example, gray mold disease resulting from *Botrytis cinerea* in the postharvest stage can easily destroy the appearance of the strawberry fruits and decrease the accumulation of total soluble solids, vitamin C, and anthocyanins, leading to severe damage to the strawberry fruit's storage capacity (*Yu et al., 2021*). Soilborne pathogens *Fusarium oxysporum* and *Macrophomina phaseolina* were detected to evidently modify strawberry fruit aroma by altering the levels of specific volatile compounds with an important impact on fruit quality (*Pastrana et al., 2023*). Studies have also shown that the contents of anthocyanins, ascorbic acid, and malic acid in strawberry fruits may have an important relationship with their sensitivity to *Botrytis cinerea* (*Li et al., 2022*). Meanwhile, some pests will also attack strawberry plants and fruits, transmitting viruses, and lead to the decline of strawberry quality. The strawberry aphid, *Chaetospihon fragaefolii*, can weaken the growth vitality of strawberry plants by feeding on plant sap, thereby affecting fruit development and overall yield (*Van Oystaeyen et al., 2022*). It should be pointed out that strawberry fruits are prone to various diseases and pests at different developmental stages, even during postharvest storage and transport, causing irreversible losses to the formation of strawberry quality. Future research should focus more on the research and development of green prevention and control measures of strawberry diseases and pests, and provide technical support for disease-free and high-quality strawberry production.

## Abiotic stress
### Temperature

Suitable temperature is an important factor for promoting strawberry plant growth and fruit quality formation over the period of growth process. A previous study indicated that a temperature difference of 12 °C (31 °C/19 °C, day and night temperature) is the optimal condition for strawberry growth and sugar accumulation (*Wu et al., 2021*). If the temperature is maintained within an ideal range, it will accelerate the growth rate of strawberry plants and improve fruit quality, such as the firmness of fruit. Low temperature

inhibited anthocyanin accumulation and led to poor coloration of strawberry fruits, thus largely making difference on their commercial value (*Mao et al., 2022*), whereas high temperature could alter fruit shape and color, as well as nutritional quality, such as sugars, acids, and phenolics (*Shirdel et al., 2025*; *Ullah et al., 2024*). It is necessary to strengthen the research on the mechanism of low/high temperature tolerance of strawberry and speed up the cultivation of new varieties to adapt to the era of increasingly fierce climate change.

## Water stress

Water stress caused by water deficit or excess is the core problem restricting the sustainable development of agriculture, seriously hindering crop growth and development, and ultimately affecting the yield and quality. In a previous study, water deficit reduced volatile ester contents, negatively affecting aroma formation in strawberry (*Rodríguez-Arriaza et al., 2025*). By contrast, the moderate reduction in water supply resulted in a rise in the final fruit brightness in all strawberry genotypes, accompanied by a hardening of the fruit firmness and sweetness (*Raffaelli et al., 2025*). Similarly, water deficit enhanced the accumulation of γ-tocopherol in strawberry fruits over the period of ripening (*Casadesús et al., 2020*). An experiment with four different cultivars indicated that water stress promoted all biochemical features in strawberry fruits such as total phenol, total anthocyanin, antioxidant activity, and sugar contents (*Adak, Gubbuk & Tetik, 2018*). These results revealed that appropriate drought stress treatment at some stages could induce the accumulation of nutrients in strawberry fruits.

## Salt stress

Strawberry is one of the horticultural crops sensitive to salt stress. Severe salt stress dramatically reduces root vitality and disrupts water and nutrient uptake, thereby inhibiting their growth, development, yield, and overall quality. When compared with the control group, the fruit firmness and accumulation of soluble solids, titratable acidity, ascorbic acid, and total phenolic compounds in strawberry fruits markedly decreased with increasing salinity (*Zahedi et al., 2020*). However, in a separate survey, the content of phenylpropanoids and ascorbic acid was increased in the strawberry fruits exposed to salt stress (*Crizel et al., 2020*). Similarly, strawberries acquired under salinity treatments recorded higher soluble solids content and enhanced fruit taste with higher contents of antioxidants compounds (*Cardeñosa et al., 2015*). Therefore, the different responses of strawberry fruits to salt stress may depend on the duration and degree of stress, the experimental system used, and the development stage of strawberry plants.

## Heavy metal

Heavy metal pollution is one of the major threatening factors to the safe production of agricultural products, posing potential risks to both food security and human health. When grown in soil with cadmium pollution, strawberry plants underwent various physiological and biochemical changes, which directly affected their nutrient uptake, ion balance, yield and fruit quality formation (*Dogan et al., 2022*). The accumulation of heavy metals such as lead and cadmium in the soil is absorbed by strawberry root systems and enriched in fruits, which affects the nutritional composition and flavor of the fruits (*Yang et al., 2022*). In

general, the harm of heavy metals is not just to affect the growth and quality of strawberry, but more importantly, eating strawberries contaminated with heavy metals will cause a series of health problems, which are not easy to be detected initially (*Rachappanavar et al., 2024*).

## Exogenous substances
### Auxin

Auxin is well recognized as a hormone essentially required for fruit growth from flower formation to fruit ripening. Auxin treatment affected the activity of cell wall remodeling enzymes and delayed the ripening process of strawberry fruits, contributing to fruit structural integrity and extended shelf life (*Castro et al., 2021*). Similarly, auxins were demonstrated to function in the receptacle fruit development and, at the same time, suppressing ripening by inhibiting related genes (*Medina-Puche et al., 2016*). At an early developmental stage, higher levels of auxin were detected to induce *AUXIN RESPONSE FACTOR 2* (*FaARF2*) expression, inhibiting *9-CIS-EPOXYCAROT-ENOID DIOXYGENASE 1* (*FaNCED1*) transcription, ABA accumulation, and suppression of receptacle ripening. During the later development stages, the decline in auxin content and the increase in ABA resulted in a decrease in *FaARF2* expression in the receptacle, which stimulated *FaNCED1* expression and ABA biosynthesis to accelerate ripening (*Li et al., 2024a*). These results indicated that the ripening process of the receptacle (pseudocarp), the main edible part of strawberry fruit, is suppressed by the high level of auxin during early development.

### Gibberellin

Bioactive gibberellins (GAs) were demonstrated to alter their accumulation during strawberry fruit development, with the highest level at the white stage, suggesting an important role of GAs in strawberry fruit (*Csukasi et al., 2011*). GA induced reduced malformed and button berries and higher marketable fruit yield without adverse effect on fruit quality (*Sharma & Singh, 2009*). However, in another study, when exposed to exogenous $GA_3$, the process of strawberry fruit ripening was slightly delayed (*Gu et al., 2019*). Application of $GA_3$ at the bottom stage showed a greater decrease in organic acids, such as citric acid, tartaric acid, and malic acid in strawberry fruits (*Taş et al., 2021*). Similarly, the organic acid and individual phenolic compound composition of strawberry fruit could be modified by $GA_3$ application (*Gundogdu et al., 2021*). As mentioned above, GA could not only regulate the development of strawberry fruit, but also induce the changes of strawberry quality parameters.

### Jasmonic acid

Jasmonic acid plays an important role in regulating the production of bioactive substances and improving fruit quality in plants (*Wu et al., 2024*). Compared to the control group, methyl jasmonate (MeJA) treatment evidently promoted the total phenolic compound, flavonoid, gallic acid, and pectin contents in strawberry fruits, whereas lower cellulose content was observed after MeJA application (*Zhang et al., 2022b*). Also, MeJA treatment increased ascorbic acid content and anthocyanin accumulation during post-harvest storage,

whereas no obvious variation on fruit firmness and lignin content was observed (*Zuñiga et al., 2020*). When exposed to MeJA after harvest, the strawberry fruits displayed higher levels of total acids, anthocyanins, total phenolics, and antioxidants, as well as lower decay incidence rate, substantially extending the fruit postharvest life (*Vaezi et al., 2022*). Reasonable use of jasmonic acid can not only improve the flavor and quality of strawberry fruit, but also significantly prolong the shelf life of postharvest strawberry.

### Ethylene

Ethylene is a commonly used plant growth regulator, mainly utilized to promote fruit ripening and improve quality. When exposed to exogenous ethephon, the strawberry fruits showed increased soluble solids and anthocyanin content, reduced hardness, and decreased organic acid content (*Yu et al., 2024*). Similarly, the firmness, anthocyanins, amino acids, and volatiles of strawberry fruits were affected after ethephon application; however, these alterations hinged on fruit developmental stage at which the treatment was performed (*Reis et al., 2020*). Continuous exposure to ethylene induced sucrose hydrolysis, malic acid catabolism, and accumulation of phenolics in cold stored strawberry fruits (*Tosetti et al., 2020*). Overall, proper application of ethephon can effectively improve the appearance, flavor, texture, and nutrients of strawberry fruits, but at the same time, attention should be paid to prevent strawberry from being too ripening and causing quality degradation.

### Cytokinin

Cytokinin is a type of plant hormone that has been shown to significantly influence fruit quality in strawberry. When treated with BA, strawberry plants displayed an evident increase in the number of flowers and fruit size or weight (*Pérez-Rojas et al., 2023*). Exogenous kinetin (KT) and 6-benzylaminopurine (BA) were found to increase accumulation of soluble sugar, anthocyanin, ascorbic acid, and total phenolics in strawberry fruits, especially at the later stages of fruit development (*Dong et al., 2021*). Application of exogenous cytokinin, forchlorfenuron (CPPU), induced total soluble solid (TSS), total acidity, and ascorbic acid production, whereas the volatile biosynthesis was suppressed in strawberry fruits (*Li et al., 2016*). As mentioned above, it was found that cytokinin could not only regulate the flower development and fruit size in strawberry, but also induce the accumulation of related secondary metabolites.

### Polyamines

Polyamines, widely existing in animals and plants, are a kind of aliphatic nitrogen-containing bases. The most common polyamines in plants are putrescine (Put), spermidine (Spd), and spermine (Spm). A recent study indicated that altered polyamine homeostasis resulted in changes in strawberry plant development, fertility, and fruit ripening (*Huang et al., 2025*). Exogenous Spm and Spd enhanced anthocyanin accumulation and coloration in strawberry fruits, whereas an opposite effect was observed in strawberry fruits exposed to Put (*Guo et al., 2018*). Applications of Spm and Spd increased titratable acids and vitamin C contents, phenolic compounds, and antioxidant activity during strawberry fruit storage, whereas the rate of fruit decay and soluble solids content were inhibited compared to the control group, increasing the shelf life of strawberries and maintaining their

nutritional values (*Jalali et al., 2023*). These results indicated that polyamines participate in the synthesis and accumulation of sugars, acids, and volatile compounds during the ripening and storage process of strawberries.

## Gamma-aminobutyric acid

Gamma aminobutyric acid (GABA), a four-carbon and non-proteinogenic amino acid, is recognized as a safe compound to regulate quality formation during fruit development and maintain postharvest quality during storage. GABA application led to higher accumulation of total soluble sugar, titratable acid, total anthocyanins, and total flavonoids, contributing to the improved postharvest marketability of strawberry (*Zhang et al., 2024*). Similarly, GABA was demonstrated to be involved in the process of extending fruit postharvest life, enhancing phytochemical compounds, and decreasing decay incidence rate regulated by MeJA (*Vaezi et al., 2022*). Preharvest application of GABA at red-turning stage dramatically improved fruit color, firmness, soluble solids, citric acid, and soluble sugars contents at harvest, whereas inhibited decline of organic acids, soluble sugars, firmness, color, and soluble solids was observed after GABA treatment at postharvest stage (*Zheng et al., 2025*). In summary, the application of GABA provides an effective means for the storage of postharvest strawberries, which can maintain fruit quality and prolong market shelf life.

### Melatonin

Melatonin (N-acetyl-5-methoxytryptamine), an antioxidant and signal molecule, can prevent adverse effects triggered by stress and improve plant growth and quality formation in plants. The injection of melatonin at the light green stage resulted in a lower total soluble solid content and fruit color, but increased titratable acidity, total phenol content, and softening (*Mansouri, Koushesh Saba & Sarikhani, 2023*). Exogenous melatonin at the light green stage could enhance the accumulation of phenols and anthocyanins, thus accelerating the process of strawberry fruit ripening (*Mansouri et al., 2021*). Melatonin effectively maintained strawberry fruit brightness, color, firmness, titratable acidity, and accumulation of total soluble solids, thereby ensuring the postharvest quality of strawberries (*Promyou, Raruang & Chen, 2023*). Similarly, melatonin application could significantly reduce deterioration rate, maintaining strawberry fruit firmness and repressing darkening (*Arabia, Muñoz & Munné-Bosch, 2025*). Overall, melatonin has shown positive effects on improving strawberry fruit quality by regulating reactive oxygen species production and maintaining the homeostasis of quality attributes.

### Polysaccharides

In recent years, the natural polysaccharides isolated from organisms have been extensively used in food and pharmaceutical industries due to their biological activities and pro-antioxidant defense capacity with few adverse effects. Strawberry fruits coated with polysaccharide exhibited marked delay in decay rate and higher concentrations of soluble solids, anthocyanins, and vitamin C as compared to the control. In addition, polysaccharide coating increased the 2,2-diphenyl-1-picrylhydrazyl (DPPH) radical scavenging activity of the fruits to improve the shelf-life of strawberries (*Yuan et al., 2020*). Similarly, polysaccharide coatings maintained higher ascorbic acid and total phenolic contents,

and significantly inhibited the softening process and decay rate of strawberry fruits (*Li et al., 2017*). Significant improvements in strawberry fruit performance were observed in terms of changes in titratable acidity, total soluble solids, total phenolic content, and ascorbic acid content by hyaluronic acid–based edible polysaccharide-protein coatings, extending the shelf-life of strawberry fruits (*Al-Hilifi et al., 2024*). Overall, polysaccharide coating has significant effects on maintaining postharvest quality and prolonging shelf life of strawberry fruits.

## OMICS IN STRAWBERRY QUALITY

### Genomics

Genomics can help to analyze the structure, function, and evolution of strawberry genome, systematically revealing the key genetic mechanism underlying fruit quality formation and providing a theoretical basis for molecular breeding. A near-complete chromosome-scale assembly for cultivated octoploid strawberry revealed that the dominant subgenome might largely contribute to strawberry quality attributes such as flavor, color, and aroma (*Edger et al., 2019*). The haplotype-resolved genome assembly of cultivated octoploid strawberry identified genes related to the anthocyanin biosynthesis pathway, revealing the structural diversity and complexity in the expression of the alleles in the octoploid strawberry genome (*Mao et al., 2023*). In a previous study, the phased genome assemblies of a highly-flavored breeding selection combined with multi-omics datasets identified candidate functional alleles affecting strawberry flavor formation (*Fan et al., 2022b*). In addition, the current genomes of strawberry combined with quantitative trait locus (QTL) mapping have largely accelerated the exploitation of quality-related candidates. A diversity panel enriched with unique European accessions identified strong candidates for fruit color, firmness, sugar and acid composition, glossiness, and skin resistance (*Prohaska et al., 2024*). Environmentally stable QTLs for strawberry ascorbic acid content were observed using an $F_1$ population originating from parental lines 'Candonga' and 'Senga Sengana' (*Muñoz et al., 2023*). Also, large-scale genome-wide association studies with two populations totaling 3,399 individuals identified two stable QTLs on chromosome 3B and 6A for soluble solids content (*Fan et al., 2023*). Genomics can decipher the genetic information of strawberry, not only revealing the molecular basis of quality traits such as sugar and acid metabolism, color formation, aroma synthesis, but also promoting the application of molecular marker breeding and gene editing technology. In the future, with the improvement of genome annotation accuracy and multi omics integration, strawberry quality improvement will be more efficient and accurate.

### Transcriptomics

Transcriptome technology employs high-throughput sequencing to analyze gene expression profiles, unveiling key regulatory mechanisms in strawberry fruit development, ripening, and stress responses. It identifies differentially expressed genes influencing sugar accumulation, anthocyanin synthesis, and aroma formation, thereby elucidating secondary metabolic pathways and providing molecular targets for quality improvement. Gene transcripts generated by transcriptome data revealed that exogenous arginine affected

expression of genes responsible for firmness, anthocyanin content, sugar content, and ethylene emissions to inhibit strawberry fruit coloration and ripening (*Lv et al., 2020*). Transcriptome sequencing revealed that the stimulation of the phenylpropanoid biosynthesis and alpha-linolenic acid metabolic pathways accounted for the formation of aroma compounds in creamy strawberry cultivars (*Fang et al., 2024*). Transcriptome data of developing strawberry fruits identified long non-coding RNAs (LncRNAs) that were associated with lipid metabolism, organic acid metabolism, and phenylpropanoid metabolism, indicating an important role of LncRNAs in strawberry fruit ripening (*Chen et al., 2022*). Also, lncRNAs related to anthocyanins in strawberries were generated by transcriptome, providing new insights into the anthocyanin regulatory network (*Lin et al., 2018*). Comparative transcriptome analysis identified a gene, *FaMDHAR50*, responsible for ascorbic acid accumulation, as well as formation of fruit flavor, appearance, and texture during strawberry fruit ripening (*Hou et al., 2023*). Transcriptomic analysis showed that elevated $CO_2$ significantly influenced secondary metabolic pathways related to fruit quality, delaying strawberry fruit ripening and senescence (*Li et al., 2024b*). Temperature fluctuation induced the changes in terpenoid biosynthesis, amino acid biosynthesis, and phenylpropanoid metabolism, as revealed by the transcriptome (*Zheng et al., 2022*).

## Proteomics

Proteomics can reveal the key proteins regulating sugar accumulation, color formation, flavor formation, and texture changes by systematically analyzing the protein expression dynamics in strawberry fruit development, maturity, and stress response, providing targets for molecular marker assisted breeding and postharvest preservation development to improve fruit quality. The quantitative proteomic profiling revealed that the elevated abundance of secondary biosynthetic proteins was demonstrated to be positively correlated with the accumulation of primary and secondary metabolites during strawberry fruit development (*Li et al., 2019*). Furthermore, the enzymes in the superoxide dismutase/glutathione metabolism system might play important roles in fruit ripening, which were investigated by a targeted quantitative proteomic approach (*Song et al., 2020*). A total of 12 genes involved in the lipoxygenase pathway for volatile biosynthesis showed multiple patterns in protein abundance at five developmental stages, which were performed by a large-scale untargeted proteomic study (*Lu et al., 2020*). Proteomic analysis showed that bioactive metabolites produced from *Trichoderma* affected the patterns of proteins linked with fruit quality factors, carbon/energy metabolisms, and secondary metabolism, enhancing strawberry fruit quality (*Lombardi et al., 2020*). A label-free proteomic technology detected the altered expression of proteins related to malate, sugar, ascorbate, and glutathione after $CeO_2$ nanoparticles application, providing new perspectives for understanding the mechanism of $CeO_2$ nanoparticles-induced strawberry fruit quality improvement (*Dai et al., 2022*). The effect of storage temperature on anthocyanins, aroma, and antioxidant capacity in strawberry fruits was observed from the perspective of proteomics (*Lv et al., 2022*).

## Metabolomics

Metabolomics employs high-throughput analysis to identify sugars, organic acids, phenolics, and other metabolites in strawberries, elucidating their association mechanisms with flavor, color, and nutritional properties. By comparing metabolic profile differences across varieties or cultivation conditions, it precisely pinpoints key compounds regulating sweetness and antioxidant activity, thereby guiding targeted breeding and cultivation optimization. Besides, metabolomics can help to decode stress-responsive metabolic pathways to enhance disease resistance and post-harvest preservation capabilities. Comparative metabolomic analysis of ten strawberry cultivars revealed that amino acid metabolism, anthocyanin biosynthesis, and flavonoid biosynthesis pathways may be responsible for nutrition characteristics in different strawberry species (*Wang et al., 2023a*). The concentrations of nerolidol, benzaldehyde, ethyl hexanoate, and ethyl isovalerate in the red fruit of 'Mixue' were significantly elevated compared with 'Sachinoka' strawberry, which resulted in an enhanced aroma in 'Mixue' (*Bian et al., 2022*). To further study the volatile metabolites during strawberry fruit growth and development, researchers have established a system, Automated Mass Spectral Deconvolution and Identification System (AMDIS), holding volatile metabolites from 61 strawberry cultivars (*Do et al., 2024*). With increasing maturity and storage, anthocyanins and cinnamic acid derivatives were evidently accumulated, whereas decreased catechin and condensed tannin was observed (*Kim et al., 2023*). With the continuous advancement of metabolomics technologies, there has been a surge in studies targeting specific classes of metabolites, such as flavonoids and volatile metabolites. These efforts have established a more robust foundation for in-depth research and precise improvement of strawberry quality.

## Integrated omics

Multi-omics integrated analysis systematically deciphers the molecular mechanisms underlying strawberry quality formation by consolidating genomic, transcriptomic, proteomic, and metabolomic data. For instance, correlating gene expression patterns with metabolite accumulation profiles uncovers key gene networks regulating sugar biosynthesis, aroma compound production, and color development. This multidimensional data-driven approach transcends the limitations of single-omics studies, offering robust theoretical support and technical pathways for the precise regulation of strawberry quality. Combined transcriptome and metabolomics analyses revealed that three and five genes in the sugar and anthocyanin metabolic pathways, respectively, might be the key genes mediating potassium sulfate-induced fruit quality improvement (*Zhang et al., 2025*). Integrated omics indicated that two genes, bZIP (*FvH4_2g36400*) and AP2 (*FvH4_1g21210*), contributed to the biosynthesis of flavonoids and anthocyanins in strawberry fruits under red/blue light and ultraviolet B irradiation, respectively (*Chen et al., 2024*). Similar strategy showed that a red skin color for strawberry was due to the increase in the upstream pathway of anthocyanin biosynthesis and reduction in the downstream steps in the flavonoid biosynthesis pathway (*Xiao et al., 2024*). Combined omics found that *leucoanthocyanidin reductase* and *chalcone synthase* genes were critical genes related to anthocyanins, cinnamic acid, and phenylalanine in strawberry (*Ren et al., 2024*). RNA-seq combined with $N^6$-methyladenosine (m$^6$A-seq)

indicated that methylation was indispensable for normal ripening of strawberry fruit, and MTA-mediated m6A modification enhanced mRNA stability or translation efficiency of genes in the ABA biosynthesis and signaling pathway, thereby accelerating the ripening process of strawberry fruit (*Zhou et al., 2021*).

# MODERN TECHNOLOGY IN STRAWBERRY QUALITY REGULATION

## Genetic transformation technology

Genetic transformation technology provides various innovative methods for improving the quality of strawberry fruits, especially in terms of disease resistance, flavor, and nutritional values (*Eashan & Saikat, 2024*). Overexpression of a WRKY transcription factor, FaWRKY71, presented enhanced accumulation of anthocyanin content and texture softening in strawberry fruits (*Yue et al., 2022*). The Ripening Inducing Factor (FaRIF), a NAC transcription factor highly expressed in strawberry receptacles during ripening, could regulate ripening-related processes such as fruit softening, coloration, and sugar accumulation in both FaRIF-silenced and overexpression lines (*Martín-Pizarro et al., 2021*). Overexpression of *FaRRP1*, a ripening-regulation protein involved in clathrin-mediated endocytosis of ABA, positively regulating coloring and ripening of strawberry fruits (*Li & Shen, 2023*). Reduced total sugar and anthocyanin accumulation was recorded in transgenic lines overexpressing a RING-type E3 ligase gene, *FaBRIZ*, indicating that *FaBRIZ* was a negative regulator of ripening in strawberry fruit (*Wang et al., 2023b*). Transient overexpression found that the transcription factor FaTCP7 (TEOSINTE BRANCHEN 1, CYCLOIDEA, and PCF) could bind to the promoters of two sugar transporter genes, *FaSTP13* (sugar transport protein 13) and *FaSPT* (sugar phosphate/phosphate translocator) to inhibit their transcription activities, resulting in a reduction in soluble sugar content in strawberry fruit (*Chen, Gao & Shen, 2024*). These findings show that transgenic technology has been commonly used in strawberry quality research, and more and more genes may be validated to be related to strawberry quality through transgenic technology (Fig. 1).

## Gene editing technology

Gene editing technology enables precise optimization of key quality traits such as sugar content, acidity, anthocyanin levels, and shelf life in strawberries through targeted modification of the genome. This approach overcomes the limitations of traditional breeding methods, providing an efficient strategy for developing high-quality, longer-lasting, and stress-resistant strawberry varieties. The polygalacturonase FaPG1, involved in remodelling pectins during strawberry softening, was knocked out from strawberry plants using the CRISPR/Cas9 system, resulting in improved strawberry fruit firmness and shelf life with minor changes in fruit color, soluble solids, titratable acidity, or anthocyanin content (*López-Casado et al., 2023*). Mutation of phytoene desaturase (PDS), a plant enzyme involved in carotenoid biosynthesis, in octoploid strawberry led to a clear albino phenotype at a high frequency (*Wilson et al., 2019*). When the *Reduced Anthocyanins in Petioles* (*RAP*) gene encoding a glutathione S-transferase was knocked out, the strawberry fruits showed reduced total anthocyanin contents and coloration interruption

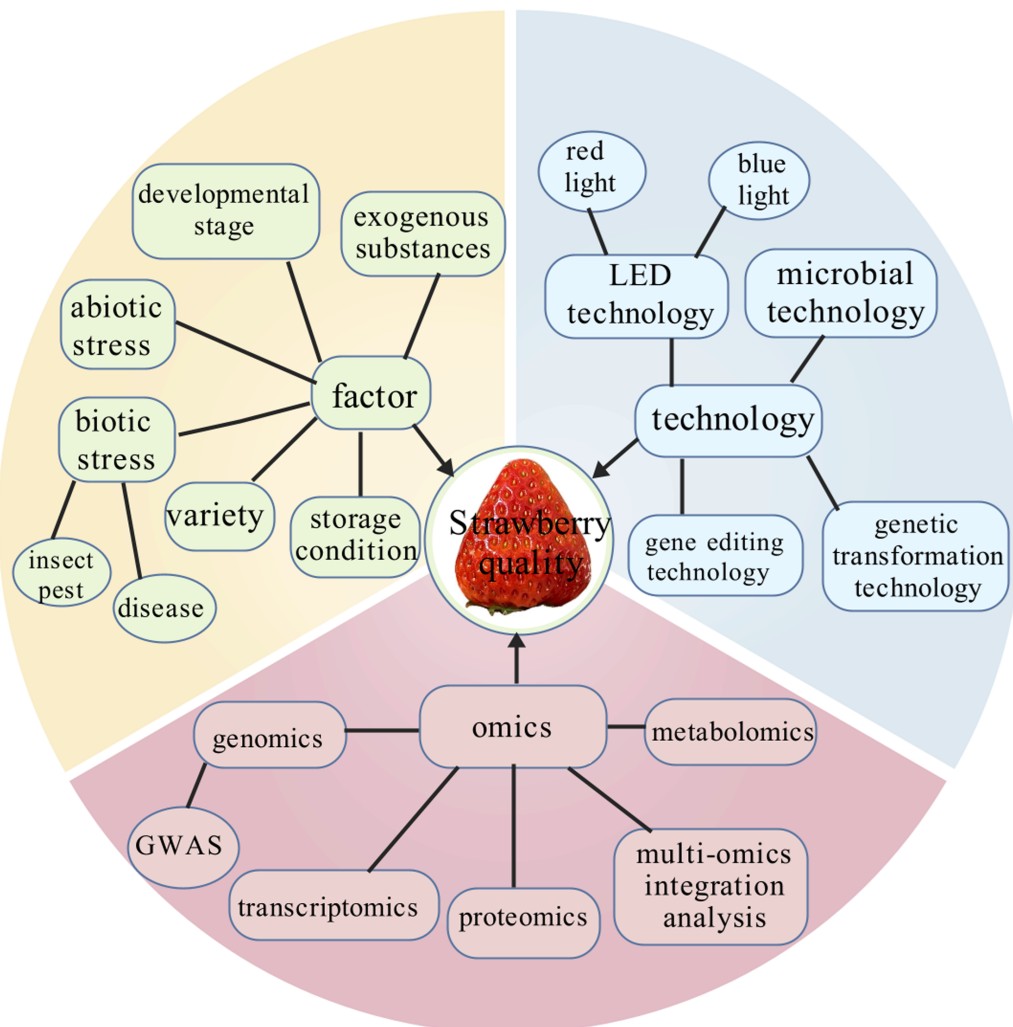

**Figure 1 Regulatory network of factors affecting strawberry fruit quality formation from omics to biotechnology.**

(*Gao et al., 2020*). This technology opens up novel opportunities for engineering strawberry and related horticultural crops to improve traits of interest. However, it should also be noted that cultivated strawberries are octoploid with a complex genome, making it challenging to precisely edit multiple alleles. Quality traits such as sugar content and firmness in strawberries are regulated by multiple genes, and single gene editing is insufficient for coordinated improvement of these characteristics.

## Light emitting diode

Light emitting diode (LED) is widely used in various stages of plant growth, especially in greenhouse cultivation and indoor agriculture, due to its high efficiency, energy saving, and adjustable spectral characteristics. Nowadays, the application of LED light technology in improving strawberry quality is receiving increasing attention. Supplemental light emitted by LED generally lowered the fruit hardness and increased the contents of soluble solids

and titratable acids in strawberry compared to the reference group (*Tang et al., 2023*). Similarly, another study also revealed that LED light could optimize flavor, nutritional value, and production of strawberries (*Hanenberg, Janse & Verkerke, 2016*). Light qualities with shorter wavelength bands enhanced total soluble solid content and modified time of fruit ripeness and fruit skin hardness, whereas longer wavelength bands promoted the growth and proliferation of strawberry plants (*Lu et al., 2024*). In addition, LED light with specific wavelengths can effectively reduce pathogens on the surface of strawberry fruits, thereby improving the storage life and market value (*Chong et al., 2022*). Therefore, LED with appropriate wavelengths and light intensity can promote plant growth, improve fruit quality, and reduce disease occurrence in strawberry.

## Microbial technology

In recent years, research on using microbial antagonists and their bioactive compounds to manage postharvest diseases in strawberry fruits has received increasing attention. This environmentally friendly approach has become one of the most promising alternatives to chemical fungicides. Application of *Debaryomyces hansenii*, a highly effective biocontrol agent, could impede the decay rate of strawberries with a higher ascorbic acid level than the control group, whereas no obvious variation in firmness, weight loss rate, soluble solids, and titratable acidity was observed (*Zhao et al., 2023*). Similarly, the edible films infused with *Bacillus subtilis* enhanced the shelf life of strawberry fruits and maintained the quality indicators, such as total soluble solids and total titratable acidity (*Torres-García et al., 2024*). In another study, strawberry fruits coated with postbiotic-based formulations, which were made up of peptide-protein extract from *Weissella cibaria* and exopolysaccharide from *Weissella confuse*, showed extended shelf-life, preserved fruit quality, and inhibited deterioration (*Tenea, Reyes & Flores, 2025*). These studies advance the comprehensive understanding of microbial-based preservation approaches for strawberries, demonstrating potential applications across various perishable products.

## PROSPECTS

### The mechanism of environmental factors regulating strawberry quality needs to be more refined

The formation of strawberry fruit quality involves complex physiological and biochemical processes, which are essentially determined by a metabolic regulatory network resulting from the synergistic effects of genetic factors and environmental elements. From a molecular biology perspective, hundreds of key genes are likely involved in regulating quality-related metabolic pathways such as sugar metabolism, anthocyanin synthesis, and volatile compound biosynthesis. The expression of these genes is precisely regulated through environmental signaling pathways. Current research is advancing toward integrated multi-omics analyses, aiming to establish a three-dimensional response model connecting environmental factors, gene expression, and metabolite accumulation. Future studies should combine high-precision environmental simulation systems with metabolic flux analysis technologies, with a focus on elucidating the molecular switch mechanisms underlying the coupling effects of multiple environmental factors.

### The intrinsic mechanism of strawberry fruit quality regulation still needs systematic analysis

The elucidation of strawberry quality regulation mechanisms requires greater clarity, fundamentally due to the incomplete understanding of its complex multi-dimensional interaction networks. Significant spatial variations in metabolic activities exist across different fruit tissues (epidermis, flesh, and pith), while gene expression profiles dynamically change throughout developmental stages (green mature stage, color transition stage, and fully ripe stage). For instance, the critical period for sugar accumulation and the peak phase for aroma synthesis exhibit temporal separation, suggesting that different quality traits are governed by independent regulatory modules. Future research should establish a holographic regulatory map of strawberry quality formation across molecular, cellular, and organ scales, thereby providing theoretical guidance for precise quality regulation in strawberry production.

### Utilizing modern technology to achieve precise regulation of strawberry quality

With the deep integration of gene editing, synthetic biology, and artificial intelligence (AI) technologies, strawberry quality regulation is entering a new era characterized by "predictability, programmability, and customizability". At the genetic level, novel gene-editing tools enable precise targeting of key genes regulating sugar metabolism and anthocyanin synthesis. By combining single-cell multi-omics data to construct dynamic metabolic network models, modular design of traits such as aroma and sweetness can be achieved. AI-driven deep learning models integrate big data from genomics, epigenomics, and environmental factors to predict the coupled effects of different environmental combinations on fruit quality. Synthetic biology technologies accelerate metabolic pathway reconstruction, realizing spatiotemporal precision induction of metabolite synthesis in strawberries. In the future, strawberry quality control will transcend traditional empirical approaches, achieving full-chain intelligentization from molecular design to cultivation management. This will provide revolutionary solutions for customized flavor strawberries and directional production of functional components.

## CONCLUSIONS

This study revealed the effects of cultivar characteristics, developmental stages, biotic and abiotic stresses, and exogenous bioactive substances on strawberry quality formation. It highlighted the application of omics in deciphering quality formation mechanisms and the driving role of biotechnological innovations in strawberry quality improvement. The article emphasized the potential of gene editing and genetic transformation for precise regulation of quality traits, while demonstrating innovative applications of emerging approaches including LED light regulation and microbiome technology in enhancing fruit nutritional value and storage characteristics. This research laid a scientific foundation

for strawberry producers and researchers to formulate targeted breeding strategies and high-quality cultivation techniques.

### Funding
This work was supported by the Scientific Research Foundation for Doctor from Huaiyin Institute of Technology (No. Z301B16531). The funders had no role in study design, data collection and analysis, decision to publish, or preparation of the manuscript.

### Grant Disclosures
The following grant information was disclosed by the authors:
Scientific Research Foundation for Doctor from Huaiyin Institute of Technology: No. Z301B16531.

### Competing Interests
The authors declare there are no competing interests.

### Author Contributions
- Zhang-Ying Wang conceived and designed the experiments, performed the experiments, prepared figures and/or tables, authored or reviewed drafts of the article, and approved the final draft.
- An-Qing Shen analyzed the data, prepared figures and/or tables, and approved the final draft.
- Yan-Xin Ge analyzed the data, authored or reviewed drafts of the article, and approved the final draft.
- Cheng-Ling Zhou performed the experiments, authored or reviewed drafts of the article, and approved the final draft.
- Yu-Shan Qiao conceived and designed the experiments, authored or reviewed drafts of the article, and approved the final draft.
- Ai-Sheng Xiong conceived and designed the experiments, authored or reviewed drafts of the article, and approved the final draft.
- Guang-Long Wang conceived and designed the experiments, prepared figures and/or tables, authored or reviewed drafts of the article, and approved the final draft.

### Data Availability
This is a literature review.

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
