# Peer review of "Regulation of fruit quality formation in strawberry: from omics to biotechnology"

_PeerJ, doi:10.7717/peerj.19497_

## Round 0.1 · original submission · Major Revisions

Please make revision according to the reviewers' comments.

Reviewer 1 ·

Basic reporting

no comment

Experimental design

no comment

Validity of the findings

no comment

Annotated reviews are not available for download in order to protect the identity of reviewers who chose to remain anonymous.

Reviewer 2 ·

Basic reporting

The manuscript ‘Regulation of fruit quality formation in strawberry: from omics to biotechnology’ written by Wang et al. introduced a lot of information about quality regulation in strawberry fruit, laying the foundation for studies aimed to promote strawberry fruit quality and industry. The intention of this article is of great significance and the content is quite comprehensive, although some points still require revision.
1. Lines 83-84: Why did the authors place this sentence here? I think that this sentence here is inappropriate.
2. I noticed that the authors used ºC to represent the degree of fruit sweetness. Is that accurate? Please confirm.
3. I have seen that there are so may diseases that can affect strawberry quality formation. The current content covers too little about the impact of pathogens on strawberry quality.
4. Lines 138-140: I haven't seen how aphids affect the quality of strawberries here. I Suggest adding more descriptions and, if possible, citing relevant literature.
5. Lines 185-186: the firmness of meat? Please check it.
6. Saline stress may also affect strawberry quality. However, in the article, the authors paid more attention to how salt stress affects the growth of strawberry plants, but there is less introduction on how salt stress affects strawberry quality.
7. Lines 246-250: I think that the content of this reference is not about the effect of heavy metals on strawberry quality.
8. Lines 279-285: I think these sentences, to some extent, deviate from the main idea that this article intends to convey.
9. To some extent, storage conditions have a significant impact on the quality changes of strawberries. However, this article did not introduce this important environmental condition. Therefore, I suggest adding this paragraph to introduce the impact of storage conditions on strawberry quality.
10. The first sentence in the conclusion part is similar to that in the abstract part, and should be revised. Also, the length of the conclusion part should be reduced by leaving out the unimportant part.
11. The format of references should be further standardized and unified to ensure a high degree of consistency in the journal name and specific details throughout the entire literature section.

Experimental design

No comments.

Validity of the findings

No comments.

Additional comments

No comments.

·

Basic reporting

The article provides a comprehensive review of the recent advances in understanding strawberry quality formation, a fruit known for its numerous nutritional benefits. The article also discusses the application of omics and biotechnology in strawberry research and quality regulation. However, there are several minor issues that should be addressed for further improvement:
1.The language of the manuscript needs to be polished to meet scientific writing standards.
2.Line 38: Please confirm whether "berry quality" is the correct term.
3.Lines 51-52: the terms "Chilean strawberry" and "Florida strawberry" are not formally recognized. It is recommended to revise this to "Fragaria chiloensis (Chilean strawberry)" and "Fragaria virginiana (Virginia strawberry, native to the eastern United States)." .
4.When discussing the application of biotechnology in strawberry quality control, the manuscript currently emphasizes the advantages of the technology. It would be beneficial to also address the limitations and challenges associated with these biotechnological approaches.
5.The "Genomics" section: it is suggested to consolidate the numerous scattered paragraphs into a more cohesive and focused section. Currently, the content appears fragmented and may be difficult for readers to follow.
6.The manuscript cites several older references. To strengthen the literature review and ensure the manuscript reflects the most current developments in the field, it is recommended to incorporate more recent studies, particularly those published in the last few years.

Experimental design

no commen

Validity of the findings

no commen

Additional comments

no commen

---

## Round 0.2 · accepted · Accept

I am please to inform you that your manuscript is acceptable for publication.

For instance:

L 48 Spelling error, should be "INTRODUCTION"

L 50 should say "It has been cultivated in Europe since the 18th century...."

Reviewer 1 ·

Basic reporting

no comment

Experimental design

no comment

Validity of the findings

no comment

Additional comments

Authors well adressed all my comments and I am satisfied with this revision.

Reviewer 2 ·

Basic reporting

The author has made corresponding revisions to the comments raised, which has greatly improved the overall quality of this manuscript. I have no further comments.

Experimental design

No comments.

Validity of the findings

No comments.

Additional comments

No comments.

·

Basic reporting

nice response.

Experimental design

no comment.

Validity of the findings

no comment.

Additional comments

no comment.